# Pilomatricoma in Syndromic Contexts: A Literature Review and a Report of a Case in Apert Syndrome

**DOI:** 10.3390/dermatopathology12030024

**Published:** 2025-08-01

**Authors:** Gianmarco Saponaro, Elisa De Paolis, Mattia Todaro, Francesca Azzuni, Giulio Gasparini, Antonio Bosso, Giuliano Ascani, Angelo Minucci, Alessandro Moro

**Affiliations:** 1Maxillo Facial Surgery Unit, Fondazione Policlinico Agostino Gemelli IRCCS Hospital, Largo Agostino Gemelli 8, 00168 Rome, Italy; gianmarco.saponaro@gmail.com (G.S.); mattia.todaro@gmail.com (M.T.); francesca.azzuni01@icatt.it (F.A.); toniobosso@gmail.com (A.B.); alessandro.moro@policlinicogemelli.it (A.M.); 2Departmental Unit of Molecular and Genomic Diagnostics, Genomics Core Facility, Gemelli Scienze and Technology Park (G-Step), Fondazione Policlinico Agostino Gemelli IRCCS Hospital, Largo Agostino Gemelli 8, 00168 Rome, Italy; angelo.minucci@policlinicogemelli.it; 3Maxillo Facial Surgery Unit, Ospedale Civile Santo Spirito Pescara, via Fonte Romana 8, 65100 Pescara, Italy; giulianoascani@gmail.com

**Keywords:** pilomatricomas, genetic syndromes, apert syndrome, pilomatrixoma, syndromes, calcifying epithelioma of Malherbe

## Abstract

Pilomatricomas are benign tumors originating from hair follicle matrix cells and represent the most common skin tumors in pediatric patients. Pilomatricomas may be associated with genetic syndromes such as myotonic dystrophy, familial adenomatous polyposis (FAP), Turner syndrome, Rubinstein–Taybi syndrome, Kabuki syndrome, and Sotos syndrome. This study reviews the literature on pilomatricomas occurring in syndromic contexts and presents a novel case linked to Apert syndrome. A systematic review was conducted using PubMed and Cochrane databases, focusing on case reports, case series, and reviews describing pilomatricomas associated with syndromes. A total of 1272 articles were initially screened; after removing duplicates and excluding articles without syndromic diagnoses or lacking sufficient data, 81 full-text articles were reviewed. Overall, 96 cases of pilomatricomas associated with genetic syndromes were identified. Reports of patients with Apert syndrome who do not develop pilomatricomas are absent in the literature. Pilomatricomas predominantly affect pediatric patients, with a slight female predominance, and are often the first manifestation of underlying genetic syndromes. Our study highlights previously unreported associations of pilomatricoma with Apert syndrome, providing molecular insights. This study contributes to understanding the clinical and molecular features of pilomatricomas in syndromic contexts and underscores the importance of genetic analysis for accurate diagnosis and management.

## 1. Introduction

Pilomatricomas are benign tumors originating from hair follicle matrix cells, located in the dermis or subcutaneous tissue [1]. Also known as “calcifying epithelioma of Malherbe”, this neoplasm was first described by Malherbe and Chenantais in 1880 and later characterized by Forbis and Helwig as “pilomatrixomas” [2]. Pilomatricomas are the second most common skin tumor in the pediatric population and account for approximately 10% of skin lesions in children [3]. Usually pilomatricomas appear as nodules with angulated edges, described clinically as the “tent sign” [4]. While generally presenting as solitary lesions, multiple pilomatricomas may develop in a minority of cases [2,5].

Although pilomatricomas are benign tumors, rare cases of malignant transformation have been reported in the literature, referred to as pilomatrical carcinoma. These malignant variants are characterized by aggressive behavior, local recurrence, and potential for metastasis. To date, only a limited number of pilomatrical carcinomas have been described, and their association with genetic syndromes remains unclear. No definitive link has been established between syndromic contexts and a higher risk of malignant transformation. However, considering the genetic instability present in certain syndromes, further studies are needed to investigate whether syndromic pilomatricomas may harbor an increased risk of oncogenic progression [6].

Pilomatricomas can be sporadic, familial (usually with autosomal dominant or recessive inheritance), or associated with genetic syndromes. The most common syndromic associations include myotonic dystrophy, familial adenomatous polyposis (FAP)-related syndromes (including Gardner syndrome), Turner syndrome, Rubinstein–Taybi syndrome, Kabuki syndrome, and Sotos syndrome [7,8].

A recent comprehensive review by Ciriacks et al. thoroughly analyzed multiple pilomatricomas in syndromic contexts, highlighting the importance of recognizing these lesions as potential early markers of underlying genetic disorders [9]. However, to date, no association with Apert syndrome has been reported.

The aim of this study is to provide an updated systematic review of pilomatricomas in syndromic contexts, adding a novel case linked to Apert syndrome.

## 2. Methods

A systematic search of the PubMed and Cochrane databases was performed to identify published case reports, case series, and retrospective reviews on pilomatricomas associated with genetic syndromes. The search terms included combinations of “pilomatricoma-associated syndromes,” “pilomatricomas and syndromes,” “multiple pilomatricoma(s),” “calcifying epithelioma(s) of Malherbe,” “syndrome(s),” and specific syndromes such as Turner syndrome, Gardner syndrome, Steinert disease, Rubinstein–Taybi syndrome, Sotos syndrome, Kabuki syndrome, and Apert syndrome. Reference lists of selected articles were also screened for additional cases. Articles written in English, Italian, Spanish, German, and Portuguese were included.

The initial search identified 1272 articles. After removing 53 duplicates and excluding 18 articles reporting only sporadic cases (including familial multiple pilomatricomas without syndrome diagnosis) and 5 articles lacking sufficient patient data, a total of 81 full-text articles were included. Cases of pilomatrical carcinoma were excluded.

This systematic review was conducted in accordance with the Preferred Reporting Items for Systematic Reviews and Meta- analyses (PRISMA) statement for reporting systematic reviews (Figure 1, Appendix A).

## 3. Case Report

On 21 April 2023, a 15-year-old female with known Apert syndrome reported a white-yellowish oval-shaped lump, in the frontal region, on her left forehead, first noticed 4 months prior to consultation.

Physical examination revealed a firm, non-tender, well-circumscribed mass, approximately 2 cm in diameter, on the left frontal bone. The overlying skin was normal in appearance, without erythema or warmth, and the mass did not transilluminate. This presentation raised the suspicion for a pilomatricoma. After biopsy and resection, on April 27th 2023, a diagnosis of pilomatricoma was confirmed.

The post-operative follow-up at 3, 6, and 12 months did not show any signs of disease recurrence.

Subsequently, a molecular analysis for associated genetic mutations was also conducted.

### Molecular Analysis

Eosin-stained histology tissue slides were examined by dedicated pathologists to identify areas of at least 20% of tumor cells content. DNA were extracted using the MagCore^®^ Genomic DNA FFPE One-Step commercial kit on the MagCore^®^ HF16 Plus II automated platform (RBC Bioscience Corp., Taiwan, China), according to manufacturer’s procedures. A comprehensive Cancer Genome Profiling (CGP) of the sample was performed using the TruSight Oncology 500 High Throughput (TSO500HT) assay. TSO500HT allows identification of Single Nucleotide Variants (SNVs), Insertions and Deletions (indels), and Copy Number Alterations (CNVs, amplifications) in 523 genes related to cancer susceptibility and treatment, along with the Tumor Mutational Burden (TMB) and Microsatellite Instability (MSI) biomarkers. The Next Generation Sequencing (NGS) was performed on the NovaSeq6000 platform (Illumina, San Diego, CA, USA). Output data evaluation was obtained using VELSERA Clinical Genomics Workspace tool, and only genomic profiling characterized by a sequencing data with a median depth of coverage > 500× was considered in the final review of molecular results. A cut-off of 5% of Variant Allele Frequency (VAF%) was adopted. Interpretation and final reporting of the detected variants were performed according to mutational databases, such as COSMIC [https://cancer.sanger.ac.uk/signatures/, last accessed on 1 February 2024]; OncoKb [https://www.oncokb.org/, last accessed 1 February 2024]; ClinVar [https://www.ncbi.nlm.nih.gov/clinvar/, last accessed 1 February 2024], as well as clinical practice guidelines and clinical trials availability. The tier classification system of the Association for Medical Pathology, the American Society of Clinical Oncology, and the College of American Pathologists was adopted [6]. Oncogenic/likely oncogenic variants classified as Tier IIC were considered of potential clinical significance for approved target therapy, practice guidelines in other tumor types, relevant evidences from multiple studies, and eligibility for clinical trials.

## 4. Result

Molecular profiling of the resected tumor revealed the presence of multiple variants classifiable as clinically relevant (Tier IIC) and not clinically relevant (Tier III). The CTNNB1 (NM_001904.3) c.110C>T, p.(Ser37Phe) oncogenic hotspot variant was identified with a VAF% of 16% (depth coverage = 927×). This variant was classified as oncogenic and of potential clinical significance (Tier IIC) based on published data supporting its role in pilomatricomas and other clinical contexts, as well as its inclusion in mutational databases such as ClinVar and OncoKb (last accessed 1 February 2024). Similarly, the FGFR2 (NM_000141.5) c.755C>G, p.(Ser252Trp) variant was identified with a Variant Allele Frequency (VAF) of 44% and high coverage depth (764×), and was considered clinically relevant based on consistent annotations in multiple databases and its known pathogenicity across various conditions. TMB was low (0.8 muts/Mb) and MSI status resulted as stable (0.9% of unstable sites, with 118/130 usable MSI sites). Multiple VUSs were also identified in the sample and considered not clinically relevant (Tier III). The overall molecular findings that emerged from the CGP are reported in Table 1.

## 5. Discussion

Mutations in CTNNB1 have been identified in a high percentage of pilomatricoma cases, and studies support their pathogenic significance, particularly in tumors associated with syndromic contexts.

The oncogenic variant identified in CTNNB1 gene c.110C>T, p.(Ser37Phe), is a missense alteration that lies within the exon 3 of the Ctnnb1 protein. Alterations occurring in CTNNB1 exon 3 and involving codons 32, 33, 34, 37, 41, and 45 disrupt the β-catenin phosphorylation sites responsible for the protein degradation and have been described as oncogenic mutations activating Wnt intracellular signaling [8,10]. In particular, amino acid residue Ser37 lies within the ubiquitination recognition motif and corresponds to a GSK3B phosphorylation site [11]. CTNNB1 p.(Ser37Phe) confers a gain of function to the Ctnnb1 protein as demonstrated by nuclear accumulation of Ctnnb1 [12,13], increased protein expression [14,15], and increased protein activity [16]. Additionally, CTNNB1 p.(Ser37Phe) was proved to cause the activation of WNT signaling and was associated with increased production of IL10 in vitro [14,17]. In ClinVar database, CTNNB1 p.(Ser37Phe) is reported as a “Pathogenic/Likely pathogenic” variant in multiple tumor types (e.g., prostate, hepatocellular, gastric, ovary, and melanoma) and as a somatic oncogenic event in pilomatrixoma (Variation ID: 17586; last accessed on 1 February 2023). This alteration was considered “likely oncogenic” in OncoKb (last accessed on 1 February 2023). CTNNB1 (also β-catenin) is a transcriptional activator involved in the WNT signaling pathway [18] with a well-documented role in different clinical contexts. In the absence of WNT ligands, β-catenin interacts with the APC/AXIN destruction complex in the cytosol. The complex includes the kinase GSK3β responsible for the phosphorylation of key β-catenin amino acidic residues and targeting β-catenin protein for cellular degradation. Engagement of WNT receptors by WNT ligands results in the disruption of the APC/AXIN complex and in the transit of β-catenin protein into the nucleus. Here, β-catenin mediates multiple target gene activations by interaction with transcription factors, including Cyclin D1 and MYC [19]. Β-catenin activation also influences cell–cell adhesion and cell migration [20]. CTNNB1 somatic mutations involving the N-terminal part of the protein have been reported in a high percentage of pilomatrixomas, approximately 75%. This frequency is greater than all the other tumors, suggested that the alteration of CTNNB1 pathway is a relevant molecular finding in pilomatrixoma [21]. From a therapeutical point of view, currently, there are no approved target therapies for tumors harboring CTNNB1 activating mutation. However, β-catenin inhibitors, such as PRI-724, are under investigation [22,23]. Additionally, Cox-2 inhibitors, including celecoxib and diclofenac, have been reported to inhibit the Wnt/β-catenin pathway and may also be relevant in tumors with β-catenin activation [24,25]. Hyperactivation of the Wnt/beta-catenin pathway may confer resistance to inhibitors of PI3K and Akt [26], and cancers with CTNNB1 mutations are presumed to be resistant also to pharmacologic inhibition of upstream components of the WNT pathway [27].

We also identified a pathogenic variant in FGFR2 gene, the p.(Ser252Trp) with a VAF% of 44.1%. This variant is reported as pathogenic germline mutation in ClinVar database with multiple submitters (Variation ID 13272, rs79184941; last accessed on 1 February 2023). In particular, this FGFR2 variant has been identified in several unrelated individuals with Apert syndrome (approximately 71%) and other FGFR2-associated craniosynostosis syndromes [28,29,30,31,32,33]. Additionally, it has been reported as an assumed de novo variant, segregating with disease in multiple families [34,35]. This variant was identified with a VAF% suggesting a germline origin, attributable to the Apert syndrome diagnosis. The FGFR2 p.(Ser252Trp) missense variant is located within the extracellular topological domain of the FGFR2 protein (aa 22-377) in the linker region between Ig-like domains 2 and 3 [36,37]. FGFR2 p.(Ser252Trp) leads to aberrant activation of MAPK signaling in vivo [28,38]. Furthermore, it increases the FGFR2 receptor binding affinity for multiple FGF ligands [39,40], increasing downstream target gene activation [41]. FGFR2 is a receptor tyrosine kinase that is a member of the fibroblast growth factor receptor (FGFR) family. Binding of FGF ligands to FGFR2 results in the rapid dimerization and activation of downstream signaling pathways including the PI3K/AKT and MAPK pathways (PMID: 28030802).

The syndromes stratified by number of pilomatricomas are presented in Table 2.

## 6. Results

The literature search of the PubMed and Cochrane databases resulted in a total of 96 cases of pilomatricomas associated with a syndrome, after excluding 8 cases of multiple familial pilomatricoma. In order to better categorize the patients, two tables were made: in the first one, the data extracted from each article was categorized for age, sex, total number of pilomatricomas, age of development of first pilomatricoma (when specified), age of syndrome diagnosis, family history of pilomatricomas, mutation of the syndrome and immunohistochemical analysis of the post-operative specimen. In the second table the data was summarized based on the scientific article, year of publication, associated syndrome, mutation, number of patients, total number of pilomatricomas and number of pilomatricomas per patient.

For cases where the number of pilomatricomas was given as a range, the lowest number was chosen for calculations to avoid falsely inflating the averages.

As shown in Table 2, 53 and other cases were associated with myotonic dystrophy, 8 and others with FAP-related syndromes (including Gardner syndrome), 14 with Turner syndrome, 11 with Rubinstein–Taybi syndrome, 1with Churg–Strauss syndrome, 2 with Kabuki syndrome, 2 with Sotos syndrome, 2 with trisomy 9, 1 with tetrasomy 9p syndrome, 1 with constitutional mismatch repair deficiency, and 0 with Apert syndrome. Sporadic cases were 1812, which were excluded from the data analysis.

There were mostly multiple pilomatricomas, even though 13 patients had single lesions.

Pilomatricomas tend to develop early in life and are more frequent in females. As a matter of fact, Table 2 reports (where there was a specification) the mean ages of onset of pilomatricomas.

A total of 53 cases of pilomatricomas associated with myotonic dystrophy were identified. Of these, 14 patients (41%) developed pilomatricomas before diagnosis of the syndrome. The average age of onset of first pilomatricoma was 27 years and 1 months, with an age range from 3 to 52. A total of 10 patients (29%) were reported to have a family history of pilomatricomas. Individuals with myotonic dystrophy and pilomatricoma exhibited an average of 8.4 ± 8.8 pilomatricomas. The number of pilomatricomas in these patients was multiple.

A total of 6 cases of multiple pilomatricomas associated with FAP-related syndromes were identified. The average age of first pilomatricoma was 9 years and 6 month, with an age range from 2 to 17.5. All 6 patients (100%) had a family history of FAP-related syndrome. All 6 patients (100%) developed 2 or more pilomatricomas prior to the identification of the FAP-related syndrome; the average number of pilomatricomas before FAP-related syndrome diagnosis was 9.2 ± 6.6. The number of pilomatricomas in these patients was multiple.

Multiple pilomatricomas, totaling 6.4 ± 4.5, were found in individuals with Turner syndrome. The average age of onset of first pilomatricoma was 15 years and 5 months, with an age range from 9 to 24. A total of 3 out of 8 patients (37.5%) developed pilomatricomas before the diagnosis of Turner syndrome; 2 of these cases had 5 or more pilomatricomas prior to the diagnosis. The number of pilomatricomas in these patients was multiple.

A total of 11 cases of multiple pilomatricomas associated with Rubinstein–Taybi syndrome were identified. The average age of onset of first pilomatricoma was 14 years and 4 months, with an age range from 5 to 20. Only 1 patient developed 15 pilomatricomas prior to the syndrome diagnosis. The number of pilomatricomas in these patients was multiple, and only in 1 patient, the number of pilomatricomas was single.

Only 2 cases of Sotos syndrome and 2 cases of Kabuki syndrome were identified. The number of pilomatricomas in these patients was multiple.

## 7. Age, Sex, and Distribution

According to the literature, pilomatricoma typically manifests in pediatric age (<18 years) in syndromic forms, although this is not an absolute rule. Generally, there are two peaks of incidence: between 0 and 20 years, and between 50 and 65 years [42]. However, the majority of cases occur before the age of 10 [43]. In the specific case of our patient, pilomatricoma was diagnosed at the age of 15. Regarding gender, the literature indicates a female predominance, and our patient was also female.

It is most commonly found on the head, neck, and upper trunk, while it rarely affects the arms and legs [3].

In the case of the patient with Apert syndrome, a single lesion of pilomatricoma was found localized in the left frontal region.

## 8. Pathophysiology and Genetic Associations

Several publications have confirmed the presence of CTNNB1 gene mutations, which encode beta-catenin, in both pilomatrixomas and pilomatrix carcinomas [44]. These mutations commonly affect the N-terminal domain of the protein and are observed in around 75% of cases. Beta-catenin is a critical component of the Wnt signaling cascade, contributing to key biological processes such as hair follicle differentiation, intercellular adhesion, and cellular proliferation [45]. Mutations, particularly in exon 3 of CTNNB1, lead to abnormal accumulation of beta-catenin within the cytoplasm and nucleus, fostering the development of tumors derived from hair matrix cells [45,46,47].

Immunohistochemical analysis has also shown increased expression of the proto-oncogene BCL-2 in basophilic cell populations within pilomatrixomas [48].

There are known links between pilomatrixoma and various inherited syndromes. For instance, in Gardner syndrome and Turner syndrome, reduced function of the APC gene results in activation of the Wnt pathway and elevated beta-catenin levels [49].

In myotonic dystrophy, mutations in the DMPK gene disrupt calcium signaling, which impairs normal epidermal proliferation and differentiation. The mutation reduces intracellular calcium availability, thereby suppressing differentiation and promoting excessive cell growth [50,51,52].

In Rubinstein–Taybi syndrome, altered CREB protein function contributes to epidermal hyperplasia. Kabuki syndrome involves mutations in the MLL2 gene, which also activate the Wnt/beta-catenin pathway [51].

Similarly, Sotos syndrome is characterized by deletions or mutations of the NSD1 gene, again leading to dysregulation of the Wnt/beta-catenin signaling route [53].

Some authors have proposed a role for PLCD1 gene mutations in cases of familial pilomatrixomas. Enhanced expression of PLCD1 (phospholipase C-d1) activates the PKC/PKD/ERK1/2 signaling axis and concurrently downregulates TPRV6, a calcium-selective channel. This mechanism stimulates keratinocyte proliferation and local calcium deposition [54].

In our reported case, the patient with Apert syndrome harbored both an oncogenic CTNNB1 variant and a pathogenic FGFR2 mutation. We also identified several variants of uncertain significance (VUS), although they were considered clinically irrelevant.

## 9. Pathological Findings

Pilomatrixomas demonstrate a progressive evolution and are histologically categorized into four stages: early, fully developed, early regressive, and late regressive phases [55].

From a cytomorphologic perspective, tumor cells are typically arranged in lobular clusters. Basaloid cells located at the periphery are actively proliferating and gradually transition into intermediate cells, eventually forming central eosinophilic ghost cells, anucleate cells with pale centers that represent lost nuclei. These ghost cells are thought to originate from primitive hair matrix keratinocytes that fail to complete their differentiation. Over time, these cells commonly undergo dystrophic calcification. Areas of focal calcification are frequently observed.

Additionally, foreign-body-type giant cells may develop as a granulomatous inflammatory reaction to keratin debris and ghost cells. The proportion of basaloid cells is typically high in early lesions, while in regressive stages, they are often scarce or absent [56,57,58].

Immunohistochemically, CTNNB1 expression is predominantly found in basaloid cells, and BCL-2 is markedly expressed in the basophilic component.

According to our literature review, no prior cases of pilomatrixoma associated with Apert syndrome have been described. Therefore, to the best of our knowledge, this report is the first documented case of a solitary pilomatrixoma occurring in a patient with Apert syndrome.

## 10. Pilomatricoma: Histological Morphological Stages

The classical literature divides pilomatricoma into four distinct evolutionary stages (Kaddu et al., 1996): early, fully developed, early regressive, and late regressive [58,59].

In the early stage, the lesion appears as a small cyst with internal fissuring, filled with keratin and ghost (shadow) cells, and surrounded by a peripheral layer of basaloid cells.In the fully developed stage, larger eosinophilic keratin masses containing numerous shadow cells are surrounded by a mantle of active peripheral basaloid cells [58].The regressive stages are marked by loss of basaloid epithelium:In the early regressive stage, residual basaloid aggregates persist at the periphery, with central shadow cells, inflammatory infiltrate, and multinucleated giant cells.In the late regressive stage, the tumor becomes an amorphous keratinized mass (often calcified/ossified), with very few viable cellular elements [58].

The following hematoxylin-eosin (HE)-stained images illustrate each stage:

Early stage (Figure 2):

At this stage, the lesion appears as a small cyst lined by squamous-basaloid epithelium. A dark layer of basaloid cells (top) transitions into a lower layer of eosinophilic, anucleate shadow cells (bottom). This corresponds to the early phase described by Kaddu et al. (1996), with immature keratin within the cyst [58].

Fully developed stage (Figure 3):

A larger, nodular tumor is observed. Compact aggregates of basaloid cells (intensely violet) surround large eosinophilic keratin shadow cell masses. This matches Kaddu’s fully developed stage, characterized by abundant shadow cells encased in a basaloid cell mantle [58].The keratin masses are broader and more consolidated than in the early stage.

Early regressive stage (Figure 4):

There is discontinuity of the epithelial layer. A cluster of basaloid cells (top) and central keratinous shadow cells (bottom, pink) are visible. Granulation tissue, inflammatory cells, and multinucleated giant cells (foreign body type) surround the lesion, indicating tumor resorption [58]. This reflects the early regressive phase described by Kaddu, with residual basaloid “islands” and inflamed shadow cells [58].

Late regressive stage (Figure 5):

The lesion now presents as an irregular, calcified mass. Numerous amorphous eosinophilic shadow cells (light pink) are interspersed with calcified/ossified thickening (bluish lamellar structures). Basaloid epithelium is almost completely absent, and inflammation is minimal. These features define Kaddu’s late stage, characterized by calcified/ossified shadow cells in a fibrous stroma [58].

## 11. Discussion

Pilomatricomas, also known as calcifying epitheliomas of Malherbe, is a noncancerous neoplasm of the skin derived from hair follicle matrix cells, typically located in the dermis or subcutaneous tissue. Our findings confirm the spectrum of pilomatricoma associations with genetic syndromes as previously outlined by Ciriacks et al. [9], who emphasized the diagnostic significance of multiple pilomatricomas as clinical markers for syndromic conditions. Despite their benign nature, understanding their associations with syndromes is crucial for early diagnosis and management.

In line with the findings of Ciriacks and colleagues, the presence of pilomatricomas should raise suspicion for underlying genetic syndromes, underscoring the importance of a multidisciplinary approach and genetic counseling [9].

Additional research is warranted to determine whether the association with Apert syndrome is merely coincidental or indicative of a broader pathogenic connection, as proposed for other syndromes in the review by Ciriacks et al. [9].

Our review focused on 96 cases of pilomatricomas, including our single case, associated with various syndromes and excluding 8 cases of multiple familial pilomatricoma, providing valuable insights into their clinical characteristics and molecular features. These findings contribute to a better understanding of pilomatricomas and their syndromic associations.

Age, Sex, and Incidence: Our analysis revealed that pilomatricomas predominantly occur in the first and second decades of life, with a relatively even distribution among other age groups. This contradicts previous suggestions of a bimodal age distribution. Additionally, we observed a slight female predominance, consistent with most studies in the literature. The overall incidence of pilomatricomas is not well studied but is estimated to be between 0.001% and 0.0031% of all dermatohistopathologic materials submitted for examination.

Syndromic Associations: Pilomatricomas can be associated with various syndromes, including myotonic dystrophy, familial adenomatous polyposis (FAP)-related syndromes, Turner syndrome, Rubinstein–Taybi syndrome, Kabuki syndrome, and Sotos syndrome. These associations may serve as early indicators of underlying syndromes, facilitating timely diagnosis and intervention. Notably, our study reported a case of pilomatricoma associated with Apert syndrome, a finding not previously documented

Molecular Analysis: Molecular profiling of pilomatricomas can provide valuable insights into their pathogenesis and potential therapeutic targets. Understanding these molecular alterations can inform targeted therapies and personalized treatment approaches.

In our case, we identified oncogenic variants in the CTNNB1 and FGFR2 genes. The CTNNB1 variant (c.110C>T, p.(Ser37Phe)) is known to activate the Wnt signaling pathway, contributing to tumorigenesis.

Therefore, in our case of pilomatricoma detected in a patient with Apert syndrome, we observed the presence of typical genetic mutations of this tumor type, particularly the mutation in the CTNNB1 gene, known to trigger the tumorigenesis process of pilomatricoma through the activation of the WNT signaling pathway. Additionally, the oncogenic variant FGF2 was identified. The FGFR2 variant (c.755C>G, p.(Ser252Trp) has been associated with craniosynostosis syndromes [28,29,30,31,32,33].

The presence of the FGF2 oncogenic variant can be explained because a group of syndromic craniosynostoses, such as Apert syndrome, has been associated with mutations in genes encoding proteins belonging to the same biological pathway, the fibroblast growth factor receptor (FGFR).

Apert syndrome is a rare genetic disease characterized by craniofacial defects and symmetric syndactyly, involving abnormal fusion of the cranial bones and fingers and toes. This condition is caused by mutations in the FGFR2 gene (a germline mutation), which is involved in regulating cell growth and development.

In fact, the abundant expression of FGFR2 is present in the cartilage of the cranial base [59,60,61,62]. In cranial sutures, FGFR2 is strongly expressed in differentiating osteoblasts and osteoprogenitor cells. Through downstream pathways, FGF/FGFR2 signaling regulates the proliferation, differentiation, and apoptosis of osteoprogenitor cells [59,63]. Early differentiation is thought to be a key factor involved in premature suture fusion [64,65,66,67,68,69].

Most cases of Apert syndrome are due to two specific point mutations in the FGFR2 gene (fibroblast growth factor receptor 2), namely a mutation c.755C>G (p.S252W) and a mutation c.758C>G (p.P253R) [28,29,30,31,32,33].

A germline mutation occurs in the germ cells, which are the cells that give rise to oocytes and sperm. These mutations can be passed on to the offspring during sexual reproduction and therefore influence the hereditary characteristics of the next generation. This type of mutation can have an impact on all the cells of the descendant’s body, as somatic cells (those that are not germ cells) develop from the mutated germ cells during embryonic development. Therefore, a germline mutation is expressed in all the cells of the organism of the individual born with the mutation. In contrast, the FGF2 oncogenic variant is a germline mutation expressed in all the somatic cells of the individual and, consequently, is also expressed in the pilomatricoma cells. On the other hand, the CTNNB1 oncogenic variant is a non-germline mutation, so it is expressed only in the pilomatricoma cells and not in all the cells of the organism. Therefore, it is likely that our case of pilomatricoma onset in a patient with Apert syndrome is a sporadic event, and, therefore, there is no association between the FGF2 oncogenic variant and pilomatricoma. However, it is important to report this case for a better understanding of the genetic and pathological correlations in these conditions.

## 12. Conclusions

Our study sheds light on the varied clinical and molecular characteristics of pilomatricomas and their associations with syndromes. Recognizing pilomatricomas early in the context of underlying syndromes is crucial for appropriate management and genetic counseling. While the case we reported in the literature, involving a patient with Apert syndrome who developed a pilomatricoma, might appear to be a sporadic occurrence, our evidence is limited, preventing us from drawing definitive conclusions. Therefore, additional studies are necessary to delve deeper into this issue. Further research is warranted to uncover the molecular mechanisms driving pilomatricoma development and to explore new therapeutic approaches.

We are left wondering whether this occurrence can truly be deemed sporadic or if Apert syndrome serves as a risk factor for pilomatricoma development. Currently, the existing literature does not provide sufficient information to definitively address these questions, underscoring the need for further investigation into this matter.

## Figures and Tables

**Figure 1 dermatopathology-12-00024-f001:**
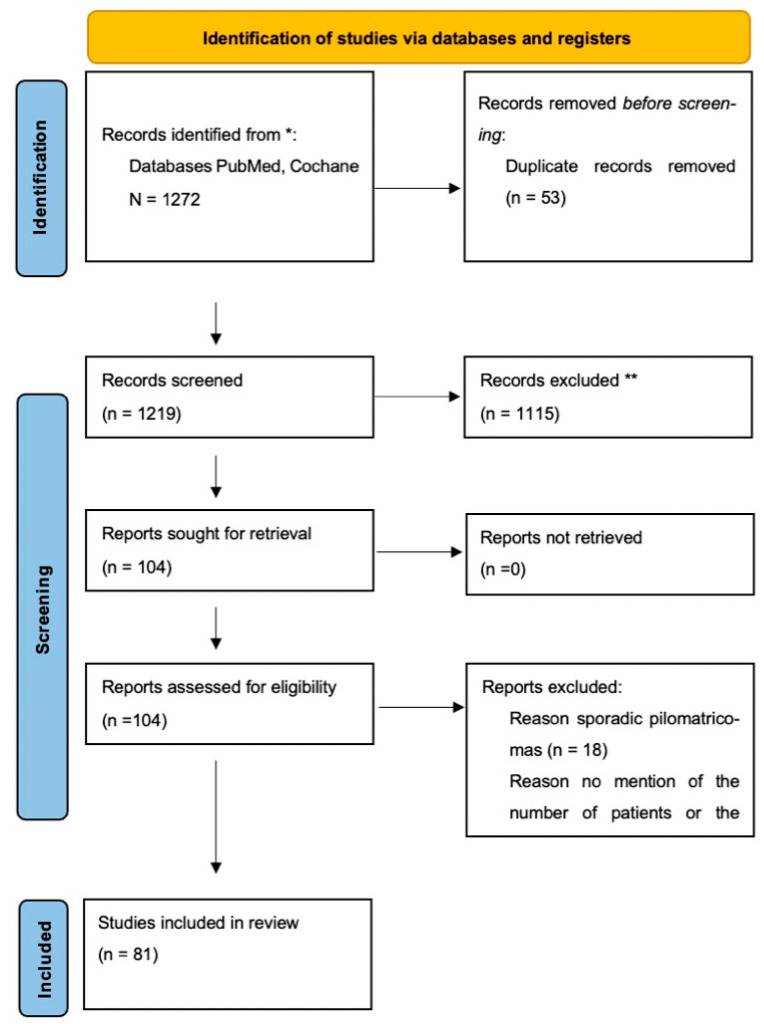
PRISMA flow diagram of the literature search and selection process. ****** Records marked with double asterisks refer to articles excluded due to irrelevance to syndromic associations or incomplete clinical information. ***** The databases searched for this review were PubMed and the Cochrane Library.

**Figure 2 dermatopathology-12-00024-f002:**
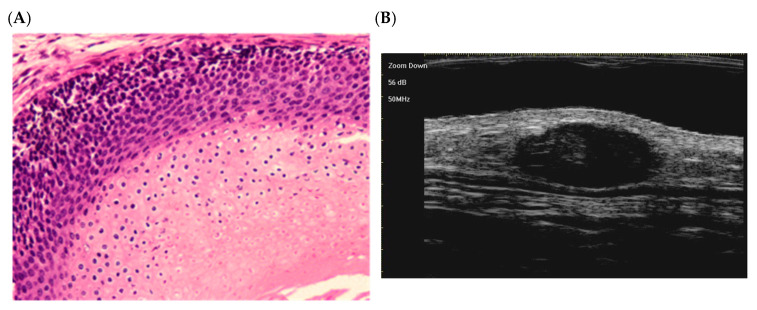
The histological presentation [(**A**) HE, original magnification ×10] and ultra-high frequency ultrasound (**B**) of PM in the early stage, indicating no obvious calcification and cystic formation [59].

**Figure 3 dermatopathology-12-00024-f003:**
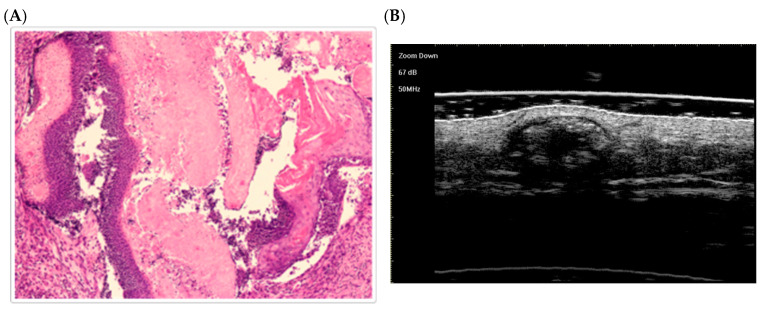
The histological presentation [(**A**) HE, original magnification ×25] and ultra-high frequency ultrasound (**B**) of PM in the fully developed stage, indicating scattered dot calcification [59].

**Figure 4 dermatopathology-12-00024-f004:**
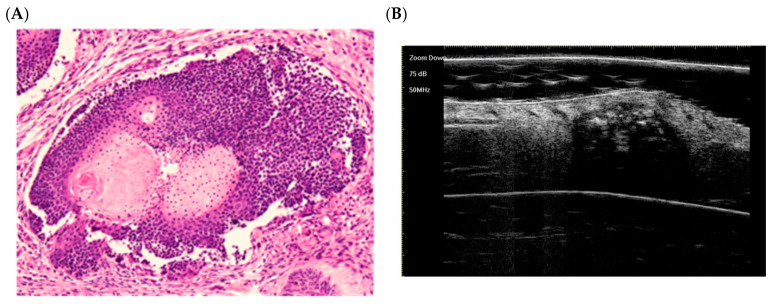
The histological presentation [(**A**) HE, original magnification ×10] and ultra-high frequency ultrasound (**B**) of PM in the early regressive stage, indicating clump calcification [59].

**Figure 5 dermatopathology-12-00024-f005:**
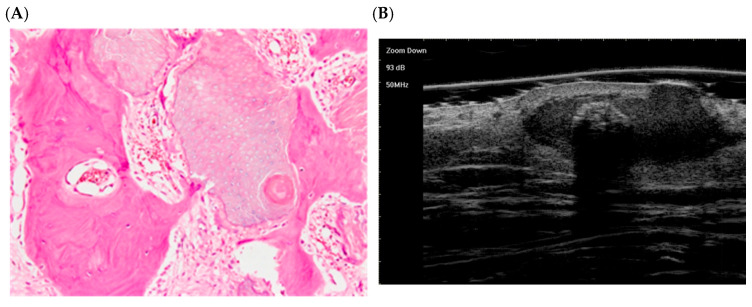
The histological presentation [(**A**) HE, original magnification ×25] and ultra-high frequency ultrasound (**B**) of PM in the early regressive stage, indicating arc calcification [59].

**Table 1 dermatopathology-12-00024-t001:** Molecular findings. The table shows the overall molecular alterations identified in the analyzed patient from the CGP analyses.

Gene	Reference Sequence	Coding Position	Amino Acid Position	VAF%	Coverage	Tier
CTNNB1	NM_001904.3	c.110C>T	p.(Ser37Phe)	16%	927X	IIC
FGFR2	NM_022970.3	c.755C>G	p.(Ser252 Trp)	44.1%	764X	IIC
CUX1	NM_181552.3	c.2507C>T	p.(Ala836Val)	49.9%	605X	III
INPP4B	NM_001101669.1	c.2017A>G	p.(Ser673Gly)	47.1%	554X	III
INSR	NM_000208.2	c.2677G>A	p.(Asp893Asn)	47.4%	426X	III
NOTCH3	NM_000435.2	c.2738C>T	p.(Pro913Leu)	50%	344X	III
PTPRD	NM_002839.3	c.68C>T	p.(Pro23Leu)	46.1%	532X	III
SLIT2	NM_004787.4	c.734A>G	p.(His245Arg)	52.3%	675X	III
SPTA1	NM_003126.4	c.4408G>A	p.(Glu1470Lys)	50.6%	712X	III
SPTA1	NM_003126.4	c.3167G>T	p.(Arg1056Leu)	47.2%	721X	III

Footnotes: VAF—Variant Allele Frequency.

**Table 2 dermatopathology-12-00024-t002:** Summary of syndromes associated with pilomatricomas: literature overview.

Syndrome	Age of Presentation	Number of Patients	Gender	Number of Pilomatricomas	Age of Onset of First Pilomatricoma	Family History of Pilomatricoma	Mutation/Pathway Activation	Immunohistochemistry
Myotonic Dystrophy	36.2 years (5–57 years)	53 and others (one article does not specify)	M/F	8.4 ± 8.8	27.1 years (3–52 years)	29%	DMPK (Myotonic Dystrophy Protein Kinase) Mutation	
FAP-related Syndromes	21.2 years (6–40 years)	8 and others (one article does not specify)	M/F	9.2 ± 6.6	9.6 years (2–17.5 years)	100%	CTNNB1 gene activation, Wnt/β-catenin pathway	CD10, β-catenin, AE1/AE3 positivity
Turner Syndrome	15.5 years (9–24 years)	14	Female	6.4 ± 4.5	5.8 years (3–8 years)		Alteration/Absence of X chromosome (karyotype 45, X)	Nuclear expression of β-catenin in basaloid cells
Rubinstein–Taybi Syndrome	14.4 years (5–20 years)	11	M/F (<F)	5.8 ± 5.2 (only 1 patient had single)	6.4 years (4–10 years)		CREBBP and EP300 mutations	
Turcot Syndrome	13.8 years (4–41 years)	NR		5.8 ± 7.7			APC gene mutation	
Kabuki Syndrome	13.8 years (4–41 years)	2		5.8 ± 7.7			MLL2 gene mutation and Wnt/β-catenin pathway activation	
21-hydroxylase Deficiency	13.8 years (4–41 years)	NR		5.8 ± 7.7				
Constitutional Mismatch Repair Deficiency (CMMRD)	13.8 years (4–41 years)	1		5.8 ± 7.7	14 years		MLH1, MSH2, MSH6, and PMS2 mutations	CTNNB1 mutation in basophilic cells
Familial Sotos Syndrome	13.8 years (4–41 years)	2		5.8 ± 7.7			NSD1 gene mutation or deletion, may also be associated with Wnt/β-catenin pathway mutation	
Apert Syndrome	15 years	1	Female (one case)	1			FGFR2 mutation	
Churg–Strauss Syndrome		1		15			Unknown	
Trisomy 9		2		Multiple			Trisomy 9	
Tetrasmy 9p Syndrome		1		Multiple (2)			Tetrasomy 9	
Familial Cases		8		2.8 ± 2.5			PLCD1 (phospholipase C-D1) upregulation, leading to increased PKC/PKD/ERK1 and 2 activity, and downregulation of the TPRV6 channel, resulting in keratinocyte proliferation and local calcium accumulation	
Sporadic Cases		1812		4.0 ± 2.8			None

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
