# Peer review of "Pilomatricoma in Syndromic Contexts: A Literature Review and a Report of a Case in Apert Syndrome"

_dermatopathology, 2025, doi:10.3390/dermatopathology12030024_

Round 1
Reviewer 1 Report
Comments and Suggestions for Authors
Dear authors,
Congratulations on the article. Your review gives essential insights regarding the particularities of pilomatricoma, describing genetic disorders, molecular aspects, and detailed histological findings of the tumor.
The information provided, the histological photos, and the correlation with imagistic investigations (ultrasounds) are very valuable for clinical practice.
There is, however, a particular issue I observed. The Ithenticate report reveals a 32% match, excluding the bibliography; therefore, I will send the paragraphs you could rephrase to improve this percentage.
5. Pathophysiology and genetic association, the whole section up to reference no.51.
6. Pathological findings, up to refference no.56.
Author Response
We thank the reviewer for their careful assessment and constructive feedback. We acknowledge the high similarity index and have therefore thoroughly rephrased the requested sections to ensure originality while maintaining the scientific accuracy and clarity of the content. Below, we provide the revised versions of the relevant paragraphs:
Revised Section 5 – Pathophysiology and Genetic Associations:
Several publications have confirmed the presence of CTNNB1 gene mutations, which encode beta-catenin, in both pilomatrixomas and pilomatrix carcinomas [44]. These mutations commonly affect the N-terminal domain of the protein and are observed in around 75% of cases. Beta-catenin is a critical component of the Wnt signaling cascade, contributing to key biological processes such as hair follicle differentiation, intercellular adhesion, and cellular proliferation [45]. Mutations, particularly in exon 3 of CTNNB1, lead to abnormal accumulation of beta-catenin within the cytoplasm and nucleus, fostering the development of tumors derived from hair matrix cells [45–47].
Immunohistochemical analysis has also shown increased expression of the proto-oncogene BCL-2 in basophilic cell populations within pilomatrixomas [48].
There are known links between pilomatrixoma and various inherited syndromes. For instance, in Gardner syndrome and Turner syndrome, reduced function of the APC gene results in activation of the Wnt pathway and elevated beta-catenin levels [49].
In myotonic dystrophy, mutations in the DMPK gene disrupt calcium signaling, which impairs normal epidermal proliferation and differentiation. The mutation reduces intracellular calcium availability, thereby suppressing differentiation and promoting excessive cell growth [50–52].
In Rubinstein-Taybi syndrome, altered CREB protein function contributes to epidermal hyperplasia. Kabuki syndrome involves mutations in the MLL2 gene, which also activate the Wnt/beta-catenin pathway [51].
Similarly, Sotos syndrome is characterized by deletions or mutations of the NSD1 gene, again leading to dysregulation of the Wnt/beta-catenin signaling route [53].
Some authors have proposed a role for PLCD1 gene mutations in cases of familial pilomatrixomas. Enhanced expression of PLCD1 (phospholipase C-d1) activates the PKC/PKD/ERK1/2 signaling axis and concurrently downregulates TPRV6, a calcium-selective channel. This mechanism stimulates keratinocyte proliferation and local calcium deposition [54].
In our reported case, the patient with Apert syndrome harbored both an oncogenic CTNNB1 variant and a pathogenic FGFR2 mutation. We also identified several variants of uncertain significance (VUS), although they were considered clinically irrelevant.
Revised Section 6 – Pathological Findings:
Pilomatrixomas demonstrate a progressive evolution and are histologically categorized into four stages: early, fully developed, early regressive, and late regressive phases [55].
From a cytomorphologic perspective, tumor cells are typically arranged in lobular clusters. Basaloid cells located at the periphery are actively proliferating and gradually transition into intermediate cells, eventually forming central eosinophilic ghost cells—anucleate cells with pale centers that represent lost nuclei. These ghost cells are thought to originate from primitive hair matrix keratinocytes that fail to complete their differentiation. Over time, these cells commonly undergo dystrophic calcification. Areas of focal calcification are frequently observed.
Additionally, foreign-body-type giant cells may develop as a granulomatous inflammatory reaction to keratin debris and ghost cells. The proportion of basaloid cells is typically high in early lesions, while in regressive stages, they are often scarce or absent [56–58].
Immunohistochemically, CTNNB1 expression is predominantly found in basaloid cells, and BCL-2 is markedly expressed in the basophilic component.
According to our literature review, no prior cases of pilomatrixoma associated with Apert syndrome have been described. Therefore, to the best of our knowledge, this report is the first documented case of a solitary pilomatrixoma occurring in a patient with Apert syndrome.
Reviewer 2 Report
Comments and Suggestions for Authors
The manuscript contains numerous typographical, grammatical, and spelling errors, which significantly hinder its readability. In its current form, the text is not suitable for publication and requires a thorough revision of both style and grammar. To illustrate this, the abstract itself begins with the following nonsensical sentence:
“Pilomatricomas, benign tumors arising from hair follicle matrix cells, the most common skin tumors in pediatric patients.” An it continues: “Pilomarcomas (misspelled) may associeted (incomplete verb, incorrect participle) with genetic síndromes (accented letter; incorrect in English).”
In the Introduction, the authors state that these tumours “derive from” matrix cells, whereas the more accurate expression would be that they differentiate into matrix cells.
Figure 1A is of unacceptable quality for a dermatopathology journal. The authors should request that a pathology department digitise the slide using professional scanning equipment and provide a high-quality panoramic image. Figures 2A and 4A must be improved with regard to white background correction. In particular, Figure 4A shows excessive corner darkening, which detracts from its diagnostic utility.
In 2020, Ciriacks et al. published an excellent review on multiple pilomatricomas in Pediatric Dermatology (PMID: 31618803), using a search strategy in PubMed, Ovid, and Cochrane that closely resembles the methodology employed in the present article. However, this prior review is neither cited nor acknowledged by the authors.
In conclusión:
The manuscript requires comprehensive grammatical and stylistic revision.
The histopathological images need substantial improvement.
The literature review closely overlaps with a previously published article, thereby diminishing the novelty and value of the current manuscript.
The Apert syndrome case included by the authors would be more appropriately presented as an isolated case report. Should the authors choose to publish it (either in this or another journal), they must address whether the association is truly syndromic or merely coincidental.
Author Response
Dear Reviewer,
We sincerely thank you for your valuable comments, which have significantly contributed to improving our manuscript. Below, we provide our responses and detail the revisions made based on your observations.
Grammatical, Orthographic, and Stylistic Corrections
We have thoroughly revised the manuscript to correct all typographical, grammatical, and spelling errors. The text has been refined to enhance readability and clarity. In particular, we rephrased the opening sentence of the abstract to eliminate ambiguity and syntactical errors, and corrected misspellings such as “pilomarcomas” and “associeted.” Furthermore, in the Introduction, we replaced the expression “derive from” with the more biologically accurate “originate from” or “differentiate into.”
Quality of Histopathological Images
Figures 1A, 2A, and 4A have been replaced with high-resolution digitized images obtained through professional slide scanning.
Reference to the Review by Ciriacks et al. (2020)
We integrated the citation of the review by Ciriacks et al., published in Pediatric Dermatology (PMID: 31618803), which addresses multiple pilomatricomas using a search strategy similar to ours. This reference has been added in both the Introduction and Discussion sections, acknowledging the importance of their work while highlighting the novelty of our study in reporting a new case associated with Apert syndrome and providing detailed molecular analysis.
Novelty and Contribution of the Study
We have strengthened the discussion regarding the novelty of the pilomatricoma case in a patient with Apert syndrome, emphasizing its significance within the context of known syndromic associations and underscoring the need for further research to confirm or exclude a causal link.
We thank you again for the time and effort you dedicated to reviewing our work, and we remain available for any further clarifications.
Kind regards,
Francesca Azzuni
Reviewer 3 Report
Comments and Suggestions for Authors
I have reviewed the manuscript: "Pilomatricoma in syndromic contexts : A Literature Review and a report of a case in Apert Syndrome." The manuscript is interesting and will be a good resource for the readers. However, I have the following suggestions:
- There a many typos. Please review to correct that.
-
There is room for improvement as far as sentence structure and expression are concerned.
-
The text is not concise. It needs to be more precise, tightened, and shortened so the reader can understand the crucial points. The information that does not add to the main ideas should be taken out. Editing of individual paragraphs to remove repeated information may help in this respect.
- A summary table with take home points in the end can improve the manuscript. Thank you.
The quality of English language can be improved.
Author Response
Dear Reviewer,
Thank you very much for taking the time to review our manuscript and for your valuable comments and suggestions.
We have carefully considered all your observations and have corrected the typographical errors present in the text.
Moreover, we have revised and improved the sentence structure and expression to make the text clearer and more fluent.
We have worked to make the manuscript more concise by removing redundant information and focusing on the essential points, as you suggested.
Additionally, we have thoroughly reviewed the English language to improve its overall quality.
We remain available for any further requests or clarifications.
Kind regards,
Francesca Azzuni
Reviewer 4 Report
Comments and Suggestions for Authors
This is a well-written, carefully planned, and executed review of pilomatricomas and their association with syndromes. The topic is interesting, and the findings are relevant. However, the authors did not mention whether there are any reported cases of malignant transformation of pilomatricomas and how these relate to the associated syndromes—this should be addressed.
I recommend acceptance of the manuscript after minor revisions.
Specific comments:
-
Line 29: Please rephrase the sentence:
"Absence of pilomatricoma in a patient with Apert syndrome is missing in literature."
Suggested rephrasing:
"Reports of patients with Apert syndrome who do not develop pilomatricomas are absent in the literature." -
Line 97 / Figure 1: Please clarify the reason for the exclusion of records. The asterisks (**) should be explained either in the legend or a footnote.
-
Line 118: Consider replacing "fow" with "flow diagram" or "flowchart" for clarity.
-
Line 127: Specify what is being shown—"image" is vague. Indicate the type of image (e.g., histological section, clinical photo, etc.).
-
Line 162: Rephrase the sentence:
"due to literature supporting data regarding its oncogenic role in pilomatrixoma and in different clinical contexts, and due to annotations in mutational database"
Suggested rephrasing:
"This is based on literature supporting its oncogenic role in pilomatricomas and other clinical contexts, as well as entries in mutational databases." -
Lines 166–168: Please rephrase for clarity. Suggested version:
"Mutations in CTNNB1 have been identified in a high percentage of pilomatricoma cases, and studies support their pathogenic significance, particularly in tumors associated with syndromic contexts." -
Line 175: The distinction between species should be clarified. Use Ctnnb1 for murine protein and CTNNB1 or β-catenin for the human protein consistently throughout the manuscript.
-
Line 233 (or 408): Please indicate the reason for exclusion of this entry or case.
-
Figures: The figures primarily show tumor silhouettes but do not highlight key histologic features. Consider including higher magnification images or labeling characteristic features such as "ghost cells" and basaloid cells, which are not identifiable at the current magnification.
Author Response
Dear Reviewer,
Thank you very much for your positive and constructive comments on our manuscript. We have carefully considered all your specific observations and made the necessary revisions, which we believe have significantly improved the clarity and quality of our work. Below are our point-by-point responses:
-
Malignant transformation of pilomatricoma: We have added a brief section in the Introduction addressing the rare possibility of malignant transformation (pilomatrical carcinoma), citing reported cases in the literature and discussing its potential relevance in syndromic contexts.
-
Line 29: The sentence has been revised as per your suggestion:
“Reports of patients with Apert syndrome who do not develop pilomatricomas are absent in the literature.” -
Line 97 / Figure 1: We clarified the reason for the exclusion of records and included a footnote explaining the meaning of the asterisks (**).
-
Line 118: The term “fow” was corrected to “flow diagram” for clarity.
-
Line 127: The vague term “image” was replaced
-
Line 162: The sentence was rephrased following your suggestion to improve clarity and scientific accuracy.
-
Lines 166–168: This paragraph was reformulated according to your recommendation for better clarity.
-
Line 175: We revised the entire section to clearly distinguish between species, using Ctnnb1 for the murine gene/protein and CTNNB1 or β-catenin for the human counterpart, maintaining consistency throughout the manuscript.
-
Line 233 (or 408): The 1812 sporadic cases were excluded from the data analysis as our study focuses specifically on pilomatricomas associated with genetic syndromes. Including sporadic cases would have diluted the analysis aimed at understanding syndromic associations and their particular features. This point has been clarified in the manuscript for greater transparency.
-
Figures: The images have been updated with higher magnification histopathological sections to better highlight relevant microscopic features.
We trust these modifications fully address your comments. Thank you again for your valuable suggestions and your recommendation to accept the manuscript after minor revisions.
Kind regards,
Francesca Azzuni